# Comparative Effects of Co-Ingesting Whey Protein and Glucose Alone and Combined on Blood Glucose, Plasma Insulin and Glucagon Concentrations in Younger and Older Men

**DOI:** 10.3390/nu14153111

**Published:** 2022-07-28

**Authors:** Avneet Oberoi, Caroline Giezenaar, Rachael S. Rigda, Kylie Lange, Michael Horowitz, Karen L. Jones, Ian Chapman, Stijn Soenen

**Affiliations:** 1Adelaide Medical School and Centre of Research Excellence in Translating Nutritional Science to Good Health, The University of Adelaide, Royal Adelaide Hospital, Adelaide, SA 5000, Australia; avneet.oberoi@adelaide.edu.au (A.O.); rsrigda17@gmail.com (R.S.R.); kylie.lange@adelaide.edu.au (K.L.); michael.horowitz@adelaide.edu.au (M.H.); karen.jones@adelaide.edu.au (K.L.J.); ian.chapman@adelaide.edu.au (I.C.); 2Food Experience and Sensory Testing (FEAST) Laboratory, School of Food & Advanced Technology, Massey University, Palmerston North 9430, New Zealand; c.giezenaar@massey.ac.nz; 3Endocrine and Metabolic Unit, Royal Adelaide Hospital, Adelaide, SA 5000, Australia; 4Faculty of Health Sciences and Medicine, Bond University, Robina, QLD 4226, Australia

**Keywords:** whey protein, dietary glucose, blood glucose, insulin, glucagon, aging

## Abstract

The ingestion of dietary protein with, or before, carbohydrate may be a useful strategy to reduce postprandial hyperglycemia, but its effect in older people, who have an increased predisposition for type 2 diabetes, has not been clarified. Blood glucose, plasma insulin and glucagon concentrations were measured for 180 min following a drink containing either glucose (120 kcal), whey-protein (120 kcal), whey-protein plus glucose (240 kcal) or control (~2 kcal) in healthy younger (*n* = 10, 29 ± 2 years; 26.1 ± 0.4 kg/m^2^) and older men (*n* = 10, 78 ± 2 years; 27.3 ± 1.4 kg/m^2^). Mixed model analysis was used. In both age groups the co-ingestion of protein with glucose (i) markedly reduced the increase in blood glucose concentrations following glucose ingestion alone (*p* < 0.001) and (ii) had a synergistic effect on the increase in insulin concentrations (*p* = 0.002). Peak insulin concentrations after protein were unaffected by ageing, whereas insulin levels after glucose were lower in older than younger men (*p* < 0.05) and peak insulin concentrations were higher after glucose than protein in younger (*p* < 0.001) but not older men. Glucagon concentrations were unaffected by age. We conclude that the ability of whey-protein to reduce carbohydrate-induced postprandial hyperglycemia is retained in older men and that protein supplementation may be a useful strategy in the prevention and management of type 2 diabetes in older people.

## 1. Introduction

Type 2 diabetes mellitus is a major and increasing problem worldwide [1,2,3]. The prevalence of diabetes increases with age; in Australia, from less than 0.3% of those under age 35 years to 14–16% in those 65 years or more [4]. Postprandial hyperglycemia is a major determinant of overall glycemic control—when HbA1C is <8% postprandial glycemic excursions are the dominant determinant of HbA1C [5,6,7,8]—and an independent risk factor for cardiovascular disease [9,10]. While ingestion of protein on its own has little effect on blood glucose concentrations [11,12], co-ingestion of dietary protein with other macronutrients may influence postprandial blood glucose concentrations when combined with carbohydrate. In non-elderly adults with and without type 2 diabetes, the oral ingestion of proteins or their component amino acids substantially reduces the increase in postprandial blood glucose concentrations when compared with carbohydrate ingestion alone [13,14,15,16,17,18], with different proteins and amino acids varying in their ability to have this effect [12,19,20,21]. Whey is a protein with potent glucose-lowering action [17,19,20,22,23,24,25] and there has been increasing interest in the possible benefits of whey protein in lowering blood glucose concentrations when ingested with a meal or as a preload [18,22].

Human ageing is characteristically associated with a loss of skeletal muscle mass and function; when severe this leads to sarcopenia and other adverse outcomes [26]. Increasing protein ingestion can preserve and increase muscle mass and function in older people and protein supplements have been shown to reduce morbidity and mortality in the elderly [26]. Whey protein is ideal for this anabolic effect as it is rich in the branched chain amino acids, particularly leucine [27]. Due to age-related anabolic resistance, more protein per serve is required to stimulate muscle protein synthesis in older than younger adults [26] and a minimum of 25–30 g protein per meal has been recommended for older people [28].

Ageing is associated with changes in glucose metabolism, including impaired glucose tolerance and reduced insulin secretion [29,30,31]. There may also be changes in the responses to dietary protein, including differing effects of both oral [11,32,33] and intraduodenal [34] whey protein administration on appetite and gut hormone release. Whey protein, when consumed orally alone, slows gastric emptying load dependently in healthy young men [35] and suppresses appetite less in healthy older men than younger men [34]. It is possible, therefore, that the capacity of whey protein to attenuate the rise in carbohydrate-induced “postprandial” blood glucose concentrations may differ between older people and young adults. Previous studies examining the effects of protein ingestion on blood glucose concentrations after carbohydrate ingestion have been conducted in young- and middle-aged adults [17,19,20,22,23,24,25]. We conducted this study to determine whether the comparative effects of whey protein/carbohydrate co-ingestion on postprandial blood glucose concentrations persists into old age. We used a whey protein dose of 30g because of its tolerability in older people and likely beneficial effects on skeletal muscle.

## 2. Materials and Methods

### 2.1. Subjects

This was a randomized double-blind cross-over study comprising 10 healthy younger men (age range 18–35 years, mean ± SEM age: 29 ± 2 years; body weight: 84 ± 4 kg; height: 1.75 ± 0.02 m; body mass index (BMI): 26.1 ± 0.4 kg/m^2^) and 10 healthy older men (68–87 years, 78 ± 2 years; 82 ± 2 kg; 1.76 ± 0.02 m; 27.3 ± 1.4 kg/m^2^). The body weight and the BMI of the younger and the older men did not differ significantly (*p* > 0.05).

The subjects were recruited by online advertisement and by flyers placed on notice boards at the University of Adelaide, Australia.

The exclusion criteria included smoking, alcohol intake of >2 standard drinks on >5 days per week, being vegetarian, intake of any illicit substance, use of prescribed or non-prescribed medications which may affect appetite, body weight, gastrointestinal function or energy metabolism, food allergy(s), known diabetes mellitus (or fasting blood glucose concentration > 6.9 mmol/L), epilepsy, gallbladder, pancreatic, known cardiovascular or respiratory disease, significant gastrointestinal symptoms, disease or surgery, any other illness deemed significant by the investigator and inability to comprehend the study protocol. Inclusion criteria included being weight stable (<5% fluctuation in weight) at study entry, as assessed by self-reported weight in the preceding 3 months, and willingness to maintain usual physical activity level throughout the study.

The Royal Adelaide Hospital Human Research Ethics Committee approved the protocol, which was conducted in accordance with the Declaration of Helsinki. The study was registered with the Australian New Zealand Clinical Trial Registry (www.anzctr.org.au (2 June 2022), registration number ACTRN12619000420145). All subjects provided written informed consent prior to their study inclusion.

### 2.2. Protocol

Each participant was studied on four occasions, separated by ~7–10 days. On each occasion, they received, in randomized order (using the method of randomly permuted blocks; www.randomization.com), a drink of either 30 g glucose (G, 120 kcal) (glucose monohydrate, Sigma-Aldrich, St. Louis, MI, USA), 30 g whey protein (P, 120 kcal) (Bulk Nutrients, Tasmania, Australia), 30 g whey protein plus 30 g glucose (GP, 240 kcal) or flavored water (control, C; ~2 kcal). The effects on blood glucose, plasma insulin, glucagon, gastric emptying, energy intake and perceptions of appetite were evaluated. The drinks were equivolemic (~250 mL) and were prepared by a research assistant who was not involved in the analysis of the study results. They were flavored with varying amounts of distilled water, sodium chloride and light lime cordial (Bickford’s Australia Pty Ltd.,Adelaide, SA, Australia) and 100 mg [^13^C ] sodium acetate to match for taste and served in a covered cup to achieve blinding. To ensure that all ingredients were dissolved evenly throughout, and to minimize the layer of foam on top of the solution, the drinks were stirred continuously at low speed on a stirring plate. Both the investigators conducting the study and the subjects were blinded to the drink composition.

Participants were told to consume the same meal on the night before each study day at around 1900 h. They were instructed to fast overnight from solids and liquids and to refrain from strenuous physical activity and alcohol for 24 h prior to their attendance at the laboratory at the Clinical Research Facility, Adelaide Health and Medical Sciences Building, the University of Adelaide at 08:30 a.m. Upon arrival, participants were seated in a chair, and a cannula was inserted in an antecubital vein for blood sampling. A heated pad was used so that the samples were arterialized. Once baseline measurements were taken (blood samples, breath sample and appetite ratings), subjects were instructed to consume the test drink within 2 min. Gastric emptying (% intragastric retention) of the drink was measured with a 13C-sodium acetate breath test *t* = 5, 10, 15, 20, 25, 30, 35, 40, 45, 50, 55, 60, 75, 90, 105, 120, 135, 150, 165 and 180 min. Perceptions of appetite were assessed with validated visual analog scales and blood samples were collected into ice-chilled EDTA-coated tubes for the measurement of blood glucose, plasma insulin and plasma glucagon concentrations at *t* = 0, 15, 30, 45, 60, 90, 120, 150 and 180 min. No inhibitors were added [33]. At 180 min, each subject was presented with a standardized cold buffet-style meal, in excess of what they were expected to consume (total energy content of 2457 kcal: 19% protein, 50% carbohydrates, 31% fat) [36], in a room by themselves to limit external distractions, and were allowed to eat freely for 30 min (180–210 min) until comfortably full.

### 2.3. Measurements

#### 2.3.1. Blood Glucose and Plasma Insulin and Glucagon Concentrations

Blood glucose concentrations (mmol/L) were quantified by the glucose oxidase method using a glucose analyser (YSI 2900 Stat Plus, Yellow Springs Instruments, Yellow Springs, OH, USA). Intra- and inter-assay coefficient of variations (CVs) were ≤ 2%. Plasma was obtained by centrifugation for 15 min, at 3200 rpm at 4 °C and samples were stored at −80 °C for further analysis of insulin and glucagon concentrations. Total plasma insulin (milliunits per liter) was measured by enzyme-linked immunosorbent assay (ELISA) immunoassay (10-1113; Mercodia, Uppsala, Sweden). The sensitivity of the assay was 1.0 mU/L and the coefficient of variation was 2.9% within and 11.6% between assays [37]. Plasma glucagon concentrations were measured by ELISA immunoassay (10-1271-01, Mercodia, Uppsala, Sweden). The lower limit of quantification (LLOQ) was 1.5 pmol/L and the detection limit was 0.75 pmol/L. The coefficient of variation was 9.3% within assays and 7.5% between assays.

#### 2.3.2. Gastric Emptying

As the stomach empties the test drink containing [^13^C] sodium acetate, the substrate is rapidly absorbed in the proximal small intestine and metabolized in the liver. ^13^CO_2_ is produced, which can then be measured in the breath, thus reflecting gastric emptying of nutrients [38]. The subjects were asked to exhale through a mouthpiece to collect an end-expiratory breath sample into a 100 mL foil bag at certain time intervals. The breath samples were collected and the quantity of ^13^CO_2_ (disintegrations per min) was measured by a non-dispersive infrared spectrometer (FANci2, Fischer ANalysen Instrumente, Leipzig, Germany) [39]. The gastric half-emptying time (T_50_) and the intragastric retention were calculated using the Wagner–Nelson method [40]. This method has been shown to be of comparable accuracy to scintigraphy in the measurement of the gastric emptying of both solid and liquid meals [41,42].

#### 2.3.3. Energy Intake

The energy consumed at the buffet meal (g) was quantified by weighing the food before and after consumption. The energy intake was calculated as absolute (kcal) at the buffet meal using commercially available software (Foodworks version 8; Xyris Software Pty Ltd, Highgate Hill, QLD, Australia) [33].

#### 2.3.4. Perceptions of Appetite 

The perceptions of hunger and fullness were rated using a 100 mm visual analog scale questionnaire immediately before the test drinks and at *t* = 15, 30, 45, 60, 90, 120, 150 and 180 min as described (29).

### 2.4. Data and Statistical Analysis

Statistical analyses were performed using SPSS software (version 25; IBM, Armonk, NY, USA). Sample size was based on the statistical power functions of the between-group contrasts of older versus younger with overall *p* = 0.05, statistical power of 0.8 and anticipated drop-out rate of ~10%, and significance levels adjusted to account for the 4 comparisons (control, whey protein, glucose, whey protein plus glucose). Calculations were performed for the primary outcome of area under the curve (AUC) of blood glucose concentrations, assuming a within-subject SD of 0.5 mmol/L, and a between-participants SD of 1.4 mmol/L [11,33,43] to detect a difference between groups of 1.5 mmol/L and a difference between treatments of 0.4 mmol/L. 

The net incremental area under the curve (Net iAUC_0–180/min_), peak, time to peak and time to return to baseline was calculated for blood glucose, insulin and glucagon. The Net iAUC_0–180/min_ was calculated from baseline using the trapezoidal rule and then divided by time (min) to present a weighted average value over the time interval. 

The main effects of age and drink condition, and their interaction effects, on blood glucose, plasma insulin and glucagon concentrations were determined using a mixed-effects model with the drink condition as the within-subject factor and age as the between-subject factor, including baseline values at each treatment visit as a covariate. Post hoc comparisons, which were adjusted for multiple comparisons with Bonferroni correction, were performed when there were significant drink-condition or interaction effects. Peak concentrations, time to peak and time to return to baseline were determined for blood glucose, plasma insulin and glucagon concentrations in those drink conditions associated with changes from baseline using a repeated measures ANOVA test (G and GP for glucose; G, P and GP for insulin and glucagon). When the time to return to baseline was between 2 blood sampling time points, an interpolated value was estimated assuming a linear relationship between the 2 time points. If a blood glucose (*n* = 5 of 80 study days) or plasma insulin (*n* = 8) or glucagon concentration (*n* = 20) did not return to baseline by 180 min, the time to return to baseline was calculated using a linear extrapolation from the values at 150–180 min. We hypothesized that insulin concentrations following both G and P drinks would increase and, to determine whether the effect on insulin of the combination drink is synergistic or additive, we compared the effect of combined whey protein and glucose (GP drink) on the rise in plasma insulin concentrations with that of the sum of the effects of glucose and whey protein alone (G drink + P drink) on insulin concentrations using a paired t-test. The inhibition of the increase in blood glucose concentrations following glucose ingestion by whey protein when compared with glucose ingestion alone was calculated as the difference between the Net iAUC of G and GP as a percentage of the Net iAUC of G. The increase in plasma insulin concentrations following GP when compared with G ingestion alone was calculated as the difference between the Net iAUC of GP and G as a percentage of the Net iAUC of G. The increase in plasma glucagon concentrations following P when compared with G ingestion alone was calculated as the difference between the Net iAUC of P and G as a percentage of the Net iAUC of G. 

Statistical significance was accepted at *p* < 0.05. Data are presented as mean values ± SEM.

## 3. Results

The study protocol was well tolerated by all subjects and no adverse effects were reported. 

### 3.1. Blood Glucose 

Baseline glucose concentrations did not differ between age groups (*p* = 0.68) or study days (*p* = 0.24, Table 1).

#### 3.1.1. Interaction Effects 

The time to peak glucose concentrations were longer after G in older than younger men (*p* = 0.03, Table 1). The age by drink interaction effect was non-significant for all other outcomes.

#### 3.1.2. Drink-Condition Effects 

Blood glucose concentrations increased from baseline following glucose ingestion, alone (G) and when combined with whey protein (GP) (both *p* < 0.05), to peak mean concentrations of 6.3–8.1 mmol/L (Table 1) but did not change following the control (C) or protein (P) drinks. 

Co-ingestion of whey protein with glucose reduced the peak and the Net iAUC blood glucose concentrations when compared with glucose ingestion alone (both *p* < 0.001), in both older and younger men, without any effect of age on this inhibitory effect (*p* > 0.05). Peak blood glucose concentrations occurred earlier after GP than G (*p* < 0.001) and returned to baseline at a comparable time after GP and G (*p* = 0.83, Figure 1). 

The inhibition of the increase in blood glucose concentrations following glucose ingestion by whey protein, when compared with glucose ingestion alone, was 44 ± 38% over 3 h (*p* < 0.001; 50 ± 13% in the first hour *p* < 0.001, 66 ± 33% in the second hour *p* = 0.001 and −1 ± 23% in the third hour *p* = 0.21).

#### 3.1.3. Age Effects 

Peak glucose concentrations after GP and G drinks were non-significantly higher in older than younger men (Table 1). Peak glucose concentrations occurred later after GP and G in the older than the younger men (*p* = 0.007) and took longer to return to baseline in the older men (*p* = 0.03). 

There was no effect of age on the inhibitory effect of whey protein (GP) on the rise in blood glucose compared with after glucose alone (G). The inhibition of the rise in blood glucose concentrations over 3 h was 62 ± 27% in the older men and 32 ± 92% in the younger men (*p* = 0.75) and 41 ± 6% in the older men and 53 ± 8% in the younger men in the first hour after drink ingestion (*p* = 0.46).

### 3.2. Plasma Insulin 

Baseline insulin concentrations did not differ between age groups (*p* = 0.22) or study days (*p* = 0.41, Table 1). 

#### 3.2.1. Interaction Effects

The peak insulin concentration was higher after G than P in younger but not older men (*p* = 0.006). Time to peak insulin was longer after P and G in older than younger men (*p* = 0.038). The age by drink-interaction effect was non-significant for all other outcomes. 

#### 3.2.2. Drink-Condition Effects 

Plasma insulin concentrations (Net iAUC) increased compared with the control following glucose (G), whey protein (P) and their combined ingestion (GP, *p* < 0.001). 

Co-ingestion of whey protein with glucose (GP) had a more than additive effect on the increase in plasma insulin concentrations (Net iAUC_0–180/min_ GP vs. G plus P: 28.3 ± 4.7 vs. 21.3 ± 3.2 mU/L∗min, *p* = 0.002, Figure 1). 

Time to peak and time to return to baseline (see age effects below) did not differ between P, G and GP (*p* > 0.05). 

#### 3.2.3. Age Effects 

Peak insulin concentrations were (non-significantly) lower after G and GP in older than younger men (*p* = 0.34). 

Peak plasma insulin concentrations occurred later in older than younger men (*p* = 0.001). The time taken to return to baseline after the caloric drinks was longer in older than younger men (*p* = 0.01). 

GP increased plasma insulin concentrations compared with G plus P; 22 ± 1% in the older men (Net iAUC_0–180min_ 27 ± 6 vs. 21 ± 5 mU/L∗min, *p* = 0.02) and 27 ± 1% in the younger men (26 ± 6 vs. 19 ± 3 mU/L ∗ min, *p* = 0.04). Peak plasma insulin concentrations were 50 ± 1% increased by GP compared with G in the older men and 41 ± 1% in the younger men (*p* < 0.001, Table 1). 

### 3.3. Plasma Glucagon 

Baseline glucagon concentrations did not differ between age groups (*p* = 0.51) or during the study days (*p* = 0.27, Table 1). 

#### 3.3.1. Interaction Effects 

The age by drink-interaction effect was non-significant for all outcomes. 

#### 3.3.2. Drink-Condition Effects 

Plasma glucagon concentrations (Net iAUC) decreased following glucose when compared with the control (*p* < 0.001), increased following combined GP (*p* < 0.001) and increased even more after whey protein alone (*p* < 0.001, Table 1, Figure 1). The increase in glucagon concentrations from baseline was greater after P than GP (*p* < 0.001). 

Whey protein increased plasma glucagon concentrations by 27 ± 1% more than G over 3 h on average (*p* < 0.001): 36 ± 1% more than G in the first hour (*p* < 0.001), 26 ± 1% more than G in the second hour (*p* < 0.05) with a comparable increase in the third hour (*p=* 0.85). 

Peak glucagon concentrations were higher after P than GP in both age groups to a similar degree (*p* < 0.001, age by drink-condition effect, *p* = 0.58, Figure 1). Time to peak/nadir after G and GP and nadir after G did not differ between treatments (*p* = 0.65). The time taken to return to baseline after the drinks was longer after P than GP or G (*p* = 0.01). 

#### 3.3.3. Age Effects 

Glucagon concentrations were not affected by age (*p* = 0.71). 

### 3.4. Gastric Emptying 

A number of data points could not be included in the analysis due to poor quality of the samples, particularly in the young men and on the control treatment day, thus the graphs/values obtained were insufficient to determine gastric emptying. Data relating to gastric emptying in these subjects were excluded from analysis. After exclusion of these data, gastric emptying data on all three protein and/or glucose drink days were evaluated in 5 younger and 10 older subjects who had data from all 3 days. 

Gastric emptying was slower after the protein (AUC_0–180/min_ gastric retention: 35 ± 2%∗min) than the glucose (30 ± 1%∗min) drink and slower after the combined glucose and protein drink (40 ± 3%∗min) than either protein or glucose alone (*p* = 0.001). Gastric emptying was non-significantly slower in older than younger men (T_50_ GP: 72 ± 13 vs. 53 ± 7 min, P: 47 ± 5 vs. 39 ± 5 min, G: 38 ± 2 vs. 33 ± 2, *p* = 0.29) (Figure 2). 

### 3.5. Energy Intake 

Ad libitum energy intake three hours following preload drink ingestion was comparable between age groups *(p* = 0.37) and drink condition (*p* = 0.95; C, P, G, GP: older: 1008 ± 89, 1053 ± 106, 1047 ± 97, 988 ± 995 kcal; younger: 994 ± 89, 1003 ± 106, 960 ± 97, 939 ± 99 kcal). For GP, the mean energy intake was lower (not significant) compared with all the drinks. 

### 3.6. Perceptions of Appetite 

Mean baseline perceptions of appetite did not differ between age groups (older vs. younger: hunger: 42 ± 9 vs. 49 ± 9 mm, *p* = 0.56; fullness: 6 ± 3 vs. 9 ± 3 mm, *p* = 0.55) or study days (*p* > 0.05). 

The protein and protein plus glucose drink suppressed hunger compared with the control during the 3 h following the drinks in the younger (AUC_0–180/min_ C, P, G, GP: 12 ± 5, 4 ± 5, 8 ± 5, 1 ± 4 mm∗min) but not the older (2 ± 5, 10 ± 5, 2 ± 5, 8 ± 4 mm∗min) men (age by drink-condition interaction *p* = 0.02, Figure 3). 

Fullness increased less following the glucose drink when compared with the other drinks in the younger (AUC_0–180/min_ C, P, G, GP: 4 ± 4, 5 ± 3, 0 ± 3, 7 ± 3 mm∗min) but not the older (7 ± 4, 6 ± 3, 11 ± 3, 8 ± 3 mm∗min) men (age by drink-condition interaction *p* = 0.01, Figure 3). 

## 4. Discussion 

The main findings of this study are: (i) the addition of 30 g of whey protein to 30 g of glucose in drink form substantially attenuated the increase in blood glucose concentrations induced by glucose alone; (ii) the magnitude of the whey-induced reduction in blood glucose was not affected by age, with comparable reductions in older to those in younger adult men; (iii) the stimulation of plasma insulin concentrations by whey protein was not reduced by ageing, unlike the insulin response to glucose; (iv) whey protein suppressed hunger less in older than younger men. 

These findings are consistent with previous observations, in which the ingestion of various proteins, including oral whey protein in doses ranging from 9 to 55 g, reduced the rise in oral carbohydrate-induced blood glucose concentration in younger adults [17,19,20,22,23,24,25]. Those findings are extended by demonstrating that these effects of whey protein are maintained into old age; all older men in this study were 68 years or older with a mean age of 78 years. Previous studies have largely included young and middle-aged adults, with none, as far as we know, studying the effects of whey protein, or other proteins, in any subjects much over 60 years. In contrast to this finding, some other responses to dietary protein are affected by ageing. Whey protein ingested alone, in doses of 30–70 g, has a reduced ability to suppress appetite and food intake in older adults compared with young adults [32,33,34] and produces greater increases in circulating concentrations of cholecystokinin (CCK) and glucose-dependent insulinotropic polypeptide (GIP) in older adults than younger adults, with comparable increases in circulating insulin [11]. Intraduodenal infusions of whey protein, at rates spanning the normal range of gastric emptying, also have different effects on feeding, gut hormone release and glucose metabolism in healthy older adults compared with young adults [34]. 

The reduced insulin concentrations and slightly higher glucose concentrations observed after ingestion of glucose alone in this study in older adults compared with younger subjects are consistent with reported changes in glucose metabolism that accompany human ageing [29,30,31]. There are varying reports as to whether the increase in circulating concentrations of the incretin hormones glucagon-like peptide-1 (GLP-1) and GIP in response to glucose ingestion is affected by ageing [44]. Indeed, GIP secretion in response to glucose ingestion may be increased in older people [11,31,44], perhaps in compensation for the age-related reduction in the ability of GIP to stimulate insulin secretion [45]. Ageing is, however, associated with islet dysfunction and impaired insulin secretion, with disordered pulsatile insulin secretion in both the fasting and hyperglycemic state [31].

Ageing-associated defects in insulin secretion have been demonstrated largely in studies where hyperglycemia has been induced by oral or intravenous glucose; they may not be present in response to protein administration. The results of the present study suggest that they are not. This would be consistent with differing mechanisms of insulin release after glucose and protein ingestion, as also would be the more than additive increases in plasma insulin after combined glucose and protein compared with either alone in this study. Nuttall et al., reported qualitatively similar at-least-additive increases in insulin concentrations after higher, 50 g, doses of protein and glucose. Ingested protein, as with glucose, promotes insulin secretion by stimulating the release of the incretins GLP-1 and GIP. The effects of increasing age on protein-induced increases in circulating incretin concentrations appear similar to those in response to glucose, with equivalent increases in GLP-1 and greater increases in GIP in older and younger subjects after whey protein ingestion [11]. 

Amino acids released by protein digestion also stimulate insulin secretion by direct action on the pancreatic islet cells [46]. In contrast to islet responses to glucose, those to amino acids may be less affected, or not at all, by ageing. Some amino acids also stimulate insulin release by acting as substrates in the Krebs cycle to create ATP, which acts directly on the beta cells to increase their insulin secretory response to glucose [46]. Support for the existence of this glucose-sensitizing effect of whey protein is provided by greater-than-additive, apparently synergistic, increases in plasma insulin concentrations after combined glucose/whey protein drinks in the present study. 

The ingestion of proteins or their component amino acids causes increases in circulating concentrations of both insulin and glucagon, with their opposing effects on blood glucose concentrations and slowing of gastric emptying [11,12], with resultant slight increases, no effect or slight decreases in blood glucose concentration when ingested on their own, depending on the type of protein or amino acid [12,20,21]. Circulating glucagon concentrations after the three drinks in this study were not affected by ageing. Glucose alone suppressed plasma glucagon concentrations equally in older and younger subjects in this study, consistent with the results of some [46,47] but not all [48] previous studies. Whey protein alone increased plasma glucagon concentrations markedly and comparably in the two age groups, as previously reported [11]. The combination whey protein/glucose drink increased glucagon concentrations equally in both age groups, with the increase approximately half that following whey protein alone. The post whey protein/glucose glucagon increase suggests a stronger stimulatory effect of whey protein on glucagon secretion than the inhibitory effect of glucose, at least with 30 g (120 kcal) of each. 

Whey is a dairy protein present in milk along with casein. Relative to casein and most other proteins, whey has higher concentrations of the amino acids leucine, isoleucine, lysine, threonine and tryptophan [49], which are among the most effective amino acids in increasing circulating insulin concentrations and reducing blood glucose when co-ingested with glucose [12,21], and has greater effects in inhibiting blood glucose increases when co-ingested with carbohydrates [19,20]. In non-elderly people with type 2 diabetes, 17 g whey protein preloads administered alone 15 min before a meal acutely reduced post-prandial glycemia and, when taken twice daily for 12 weeks combined with guar, had sustained effects to slow gastric emptying, reduce postprandial glucose and modestly reduce HbA1C [50]. Whey protein supplements are relatively low cost with few, if any, side effects in older people [26]. The finding in this study that the increase in circulating insulin concentrations and the reduction in hyperglycemia after whey protein ingestion was unaffected by ageing and persists in those over 70 years and into their 80 s, provides encouragement for further studies into the use of whey protein supplements in the prevention and treatment of hyperglycemia in older people. 

Whey protein ingestion may have beneficial effects in older people in addition to those on post-prandial glucose concentrations. There is evidence that it can lead to reductions in blood pressure, blood triglyceride levels, inflammation and oxidative stress [51], of which harmfully elevated levels are more prevalent in older adults than younger adults. Another beneficial effect of whey protein in older people is likely to be that on skeletal muscle mass and function. With increasing age there is on average a progressive loss of muscle mass and function, with approximately 5% of muscle mass lost per decade after age 30 years. When excessive, this muscle loss leads to sarcopenia, with associated substantial increases in morbidity including rates of frailty and mortality [26]. Various nutritional measures have been proposed to counteract these adverse effects of ageing, including the use of protein supplements, which has been shown to increase muscle mass and strength and may also reduce morbidity and mortality, with the greatest benefits in the most under-nourished and sarcopenic older people [26]. 

As with its effects on glucose metabolism, whey protein is a particularly good form of protein to exert this anabolic effect on muscle because of its relatively high content of branched chain amino acids, particularly leucine, which are the most effective amino acids in promoting muscle protein synthesis [27]. Some of this effect may be due to the anabolic actions on muscle of the increased insulin secretion following whey protein ingestion. Due to age-related anabolic resistance [26,52], older adults appear to require 30–35 g/serve of protein to stimulate muscle protein, compared with 20 g or less in younger adults [26], and a minimum of 25–30 g protein intake per meal for older people has been recommended by the PROT-AGE study group [28]. This provides the rationale for our use of a 30 g whey protein dose in the present study. 

Weight loss in older people, particularly if involuntary, is often associated with adverse outcomes [53] and therefore not necessarily desirable. Protein is the most satiating macronutrient in young adults, and whey protein drinks of 30–70 g suppress appetite ratings and *ad libitum* energy intake at subsequent meals in healthy young adults. Increasing age is, however, associated with a marked reduction in the satiating effects of whey protein, with no reduction in appetite ratings or *ad libitum* energy intake by older people at meals 35–510 min after 30 g whey protein drinks by older people and only minimal effects after 70 g [26]. Energy intake at the test meal in this study was not reduced significantly by any of the drinks in either age group, with a slightly greater suppression after the combined whey protein–glucose drink in the younger than the older subjects (5.5 vs. 2%). In older people, increasing the dietary content of whey protein may preserve or increase skeletal muscle at the expense of fat gain [51], without reducing appetite and energy intake. 

The longer times taken for plasma glucose and insulin concentrations to rise to peak and then return to baseline after the nutrient drinks in the older subjects than the younger subjects are probably due to a combination of slightly slower gastric emptying and delayed post-gastric effects in the older subjects. Gastric emptying of nutrient drinks was non-significantly slower in the older subjects than the younger subjects, consistent with the results of previous studies, where the slowing of gastric emptying of whey protein with age has been minor when present [26,44]. Post-gastric delays probably played a greater role. Circulating glucose and insulin concentrations increase more slowly in the older men than the younger men, even when the stomach is bypassed by infusing glucose alone [54] directly into the duodenum, and insulin concentrations increase more slowly in the older men than the younger men after intraduodenal whey protein infusion [55]. The mechanisms responsible for these post-gastric ageing-related changes are unknown. 

Our study has several limitations. The number of subjects was relatively small. Nevertheless, the results regarding glucose and insulin results were clearcut and statistically significant. We only studied men, therefore the observation may not apply to women. The results do not necessarily apply for older people ingesting doses of whey protein/glucose other than 30 g, as used in this study, although the qualitatively similar results observed with a range of protein doses in previous studies of younger subjects suggest that they will. While encouraging and suggestive, the ability of co- or pre-ingested whey protein to reduce blood glucose increases in older people after ingestion of carbohydrate in forms other than pure glucose, for example as part of a mixed meal, seems likely but will need to be established [18]. Similarly, the effects in older people of non-whey proteins and proteins in non-drink form require further study. 

## 5. Conclusions

In conclusion, 30 g of oral whey protein, a dose likely to have beneficial effects on skeletal muscle mass and function in older people, substantially attenuated the rises in blood glucose induced by 30 g of glucose alone in older men without diabetes. This extent of this reduction was not affected by ageing. The plasma insulin and glucagon responses to protein ingestion are preserved in older subjects, in contrast to the insulin response to glucose, which is reduced. These findings provide a rationale for future studies of protein supplements/preloads in older people, in whom disorders of glucose metabolism such as type 2 diabetes and of muscle metabolism such as sarcopenia are prevalent and often serious.

## Figures and Tables

**Figure 1 nutrients-14-03111-f001:**
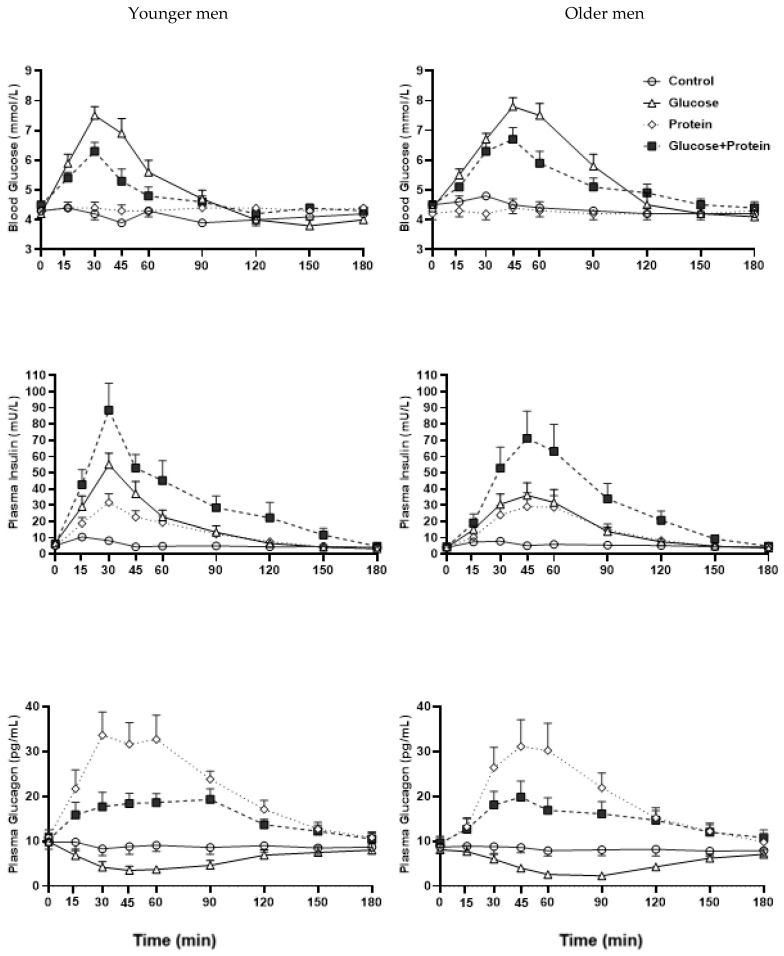
Mean (±SEM) blood glucose (mmol/L), plasma insulin (mU/L) and glucagon (pg/mL) concentrations following drink ingestion containing (i) flavored water (C, control, ~2 kcal), (ii) 30 g whey protein (P), (iii) 30 g glucose (G) or (iv) 30 g whey protein plus 30 g glucose (GP) in younger and older men. Effects of age and drink condition and the interaction effect were determined using a mixed-effect model with baseline concentrations as covariates and post hoc Bonferroni correction. T = 0 min refers to the point immediately before the drink consumption. *Age by drink-condition interaction effects*: peak insulin (*p* = 0.006) was higher after G than P in younger but not older men; time to peak glucose (*p* = 0.03) and insulin (*p* = 0.038) was longer after G in older than younger men. *Drink-condition*
*effect*: *p* < 0.001 Net iAUC glucose GP < G; *p* = 0.002 Net iAUC insulin GP > G; *p* < 0.001 peak glucagon P > GP. *Age effect*: time to peak glucose (*p* = 0.007) and insulin (*p* = 0.001) occurred later in older than younger men; time to return to baseline glucose (*p* = 0.03) and insulin (*p* = 0.01) occurred later in older than younger men.

**Figure 2 nutrients-14-03111-f002:**
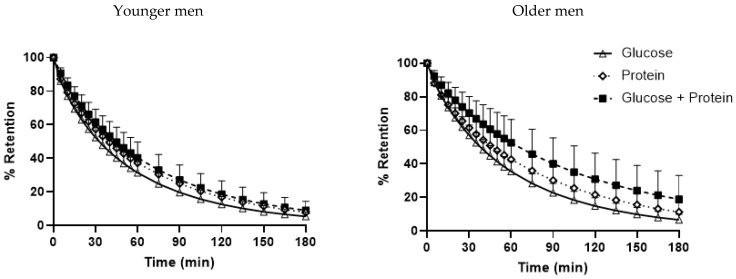
Intragastric retention (%) following drink ingestion containing (i) 30 g whey protein (P), (ii) 30 g glucose (G) or (iii) 30 g whey protein plus 30 g glucose (GP) in younger (*n* = 5) and older (*n* = 10) men. Effects of age and drink condition and the interaction effect were determined using a mixed-effect model with baseline concentrations as covariates and post hoc Bonferroni correction. T = 0 min refers to the point immediately before the drink consumption. *Drink-condition*
*effect*: *p* < 0.001 gastric emptying of GP was slower than P and both emptied slower than G.

**Figure 3 nutrients-14-03111-f003:**
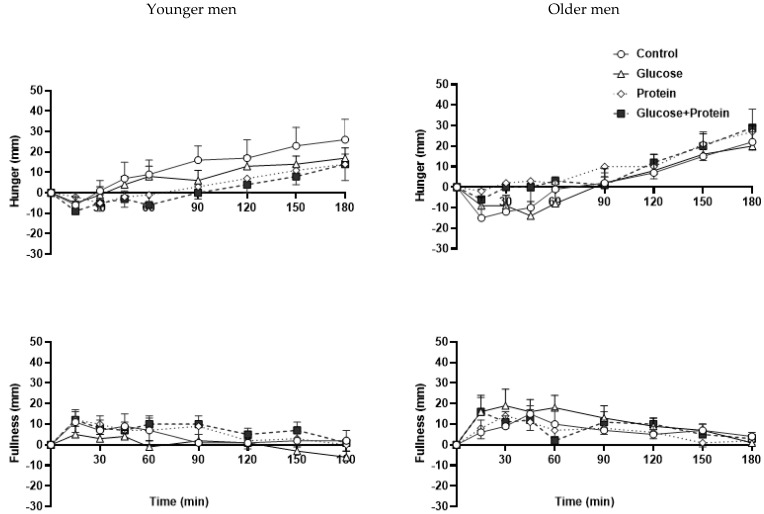
Mean ± SEM visual analogue score (VAS) of hunger (mm) and fullness (mm) following drink ingestion containing (i) flavored water (C, control, ~2 kcal), (ii) 30 g whey protein (P), (iii) 30 g glucose (G) or (iv) 30 g whey protein + 30 g glucose (GP) in younger and older men. Effects of age and drink condition and the interaction effect were determined using a mixed-effect model with baseline concentrations as covariates and post hoc Bonferroni correction. T = 0 min refers to the point immediately before the drink consumption. *Age by drink-condition interaction effects*: *p* = 0.02 hunger was suppressed by P and GP in younger but not older men; *p* = 0.01 fullness was less following G compared with the other drinks in younger but not older men.

**Table 1 nutrients-14-03111-t001:** Blood glucose, plasma insulin and plasma glucagon concentrations following control drink (C), 30 g whey protein drink (P), 30 g glucose drink (G) and 30 g whey protein plus 30 g glucose drink (GP) ingestion.

	Younger Men				Older Men				*p* Value		
	C	P	G	GP	C	P	G	GP	Age	Drink Condition	Interaction
**Blood glucose** (mmol/L)											
Fasted	4.3 ± 0.1	4.3 ± 0.2	4.2 ± 0.2	4.5 ± 0.1	4.5 ± 0.1	4.2 ± 0.2	4.4 ± 0.2	4.5 ± 0.1	0.68	0.24	0.23
Peak	4.7 ± 0.1	4.8 ± 0.1	7.8 ± 0.3	6.3 ± 0.3	4.9 ± 0.1	4.7 ± 0.1	8.1 ± 0.3	6.9 ± 0.3	0.27	<0.001	0.54
Time to peak (min)	50 ± 13	68 ± 17	34 ± 3 ^1^	30 ± 3	26 ± 13	75 ± 17	51 ± 3 ^1^	36 ± 3	0.007	<0.001	0.031
Return to baseline	88 ± 26	41 ± 13	106 ± 15	91 ± 18	60 ± 22	125 ± 23	132 ± 15	153 ± 18	0.030	0.83	0.23
Net iAUC_0–60min_	−0.1 ± 0.1	0.0 ± 0.1	2.1 ± 0.2	0.9 ± 0.2	0.1 ± 0.1	0.1 ± 0.1	2.1 ± 0.2	1.3 ± 0.2	0.29	<0.001	0.48
Net iAUC_0–180min_	−0.1 ± 0.1	0.0 ± 0.1	0.7 ± 0.2	0.3 ± 0.1	−0.1 ± 0.1	0.1 ± 0.1	1.1 ± 0.2	0.7 ± 0.2	0.08	<0.001	0.27
**Plasma insulin** (mU/L)											
Fasted	5.3 ± 1.0	4.8 ± 0.8	6.9 ± 1.1	6.2 ± 1.0	4.1 ± 1.0	4.5 ± 0.8	4.1 ± 1.1	4.5 ± 1.0	0.22	0.41	0.13
Peak	11.7 ± 1.9	33.2 ± 6.9 ^2^	55.9 ± 7.3 ^2^	93.3 ± 16.8	9.7 ± 1.9	31.7 ± 6.9	36.5 ± 7.3	72.7 ± 16.9	0.34	<0.001	0.006
Time to peak (min)	35 ± 13	32 ± 3 ^3^	33 ± 2 ^3^	36 ± 3	32 ± 4	48 ± 3 ^3^	46 ± 2 ^3^	42 ± 3	0.001	0.91	0.038
Return to baseline	73 ± 26	132 ± 8	112 ± 12	160 ± 11	109 ± 18	149 ± 8	163 ± 12	166 ± 11	0.011	0.068	0.14
Net iAUC_0–60min_	1.8 ± 0.5	16.5 ± 3.8	27.1 ± 4.4	46.2 ± 9.3	2.3 ± 0.5	15.5 ± 3.8	20.9 ± 4.4	39.8 ± 9.3	0.59	<0.001	0.26
Net iAUC_0–180min_	0.2 ± 0.6	8.3 ± 1.9	10.7 ± 2.4	25.8 ± 6.2	1.5 ± 0.6	9.6 ± 1.9	11.6 ± 2.4	26.6 ± 6.2	0.77	<0.001	0.99
**Plasma glucagon** (pg/mL)											
Fasted	9.7 ± 1.4	10.9 ± 1.6	9.8 ± 1.4	10.3 ± 1.5	8.7 ± 1.4	9.1 ± 1.6	8.1 ± 1.4	9.6 ± 1.5	0.51	0.27	0.69
Peak/Nadir	11.2 ± 1.5	36.4 ± 5.8	10.7 ± 1.4	22.6 ± 3.1	10.2 ± 1.5	33.1 ± 5.8	9.1 ± 1.4	22.1 ± 3.1	0.71	<0.001	0.73
Time to peak (min)	68 ± 22	57 ± 17	68 ± 24	54 ± 9	65 ± 22	51 ± 17	56 ± 24	48 ± 9	0.83	0.65	0.98
Return to baseline	54 ± 16	176 ± 153	71 ± 29	176 ± 51	126 ± 39	391 ± 153	133 ± 29	277 ± 51	0.11	0.011	0.69
Net iAUC_0–60min_	−0.7 ± 0.0	16.3 ± 2.6	−4.6 ± 0.7	6.3 ± 1.5	−0.1 ± 0.0	13.5 ± 2.6	−2.4 ± 0.7	6.3 ± 1.5	0.98	<0.001	0.33
Net iAUC_0–180min_	−0.9 ± 0.3	10.8 ± 1.3	−3.9 ± 0.8	5.2 ± 1.3	−0.5 ± 0.3	9.9 ± 1.3	−3.2 ± 0.8	5.1 ± 1.3	0.95	<0.001	0.92

Blood glucose (mmol/L), plasma insulin (mU/L) and glucagon (pg/mL) concentrations fasted (baseline), peak, time to peak (min), return to baseline (min), Net iAUC_0–60min_ (change from baseline area under the curve during the first hour), Net iAUC_0–180min_ (change from baseline area under the curve during the three hours) following drink ingestion containing (i) flavored water (C, control, ~2 kcal), (ii) 30 g whey protein (P), (iii) 30 g glucose (G) or (iv) 30 g whey protein + 30 g glucose (GP) in younger and older men. Effects of age and drink condition and the interaction effect were determined using a mixed-effect model using G and GP for glucose and P, G and GP for insulin and glucagon. ^1^ The age × drink-condition interaction, *p* = 0.031 (time to peak glucose is longer after G in older than younger men). ^2^ The age × drink-condition interaction, *p* = 0.006 (peak insulin concentration is higher after G than P in younger but not older men). ^3^ The age drink-condition interaction, *p* = 0.038 (time to peak insulin is longer after P and G in older than younger men). Drink-condition effects: GP vs. G reduced peak (*p* < 0.001) and Net iAUC (*p* < 0.001) glucose concentrations and the peak occurred earlier (*p* < 0.001) in both older and younger men: P, G, GP vs. C increased Net iAUC insulin concentrations (*p* < 0.001); G vs. C decreased and P, GP vs. C, P vs. GP increased Net iAUC glucagon concentrations (*p* < 0.001). Age effects: older vs. younger later peak (*p* = 0.007) and return to baseline (*p* = 0.030) glucose concentrations; older vs. younger later peak (*p* = 0.001) and return to baseline (*p* = 0.01) insulin concentrations.

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
