# Peer review of "Comparative Effects of Co-Ingesting Whey Protein and Glucose Alone and Combined on Blood Glucose, Plasma Insulin and Glucagon Concentrations in Younger and Older Men"

_nutrients, 2022, doi:10.3390/nu14153111_

Round 1
Reviewer 1 Report
In future work, I believe the authors will study women. Is there any reason to expect any difference in results in the female population?
Author Response
Reviewer 1:
In future work, I believe the authors will study women. Is there any reason to expect any difference in results in the female population?
We thank the reviewer for his/her comments. We acknowledge and have mentioned that the limitation of this study was that it was done only in men. We have previously shown that women compared to men had higher AUC blood glucose concentrations and lower plasma glucagon concentrations after 30g and 70g of whey protein drink (Giezenaar, C.; Luscombe-Marsh, N.D.; Hutchison, A.T.; Lange, K.; Hausken, T.; Jones, KL.; Horowitz, M.; Chapman, I.; Soenen, S. Effect of gender on the acute effects of whey protein ingestion on energy intake, appetite, gastric emptying and gut hormone responses in healthy young adults. Nutr Diabetes 2018, 8, 40). As men generally show greater variations in appetite and food intake in response to energy manipulation than women, and the menstrual cycle may have a confounding effect on appetite and energy intake in women, the effects of the protein drinks may be different in women and it would be appropriate to perform further studies including women (though this is not the main outcome). That said, while it would be interesting to know the effects of aging in women and we hope to do that, we have no reason to think that aging affects their responses differently to men.
Reviewer 2 Report
The authors quite soundly presented the results of a compact but well planned study on a needful topic.
A few notes.
Table 1 is not well designed. It is not clear which data refers to young and which to older men (eg GP). Worth splitting them up.
As the authors rightly point out, the samples are small, 5-10 young and 10 older men. Therefore, the effects obtained should be interpreted as trends, not regularities, and, moreover, not synergism.
However, this does not detract from the significance of the results. Congratulations on a good study! Hope to see the next investigation, including women.
Author Response
Reviewer 2:
Table 1 is not well designed. It is not clear which data refers to young and which to older men (eg GP). Worth splitting them up. As the authors rightly point out, the samples are small, 5-10 young and 10 older men. Therefore, the effects obtained should be interpreted as trends, not regularities, and, moreover, not synergism. However, this does not detract from the significance of the results.
We thank the reviewer for his/her useful and helpful comments. We have adjusted the table to more clearly separate the younger and older men results.
With regards to synergism: we have used this term in the discussion when describing the statistically significantly more than additive effect that combined glucose and whey have on circulating insulin compared to the addition of their effects when administered separately. It is our understanding that this effect of whey/glucose on insulin does therefore qualify as synergistic. As such, we would like to leave that word synergistic in the discussion, particularly as, in the first half of the sentence in which we have used it we provide a possible mechanism for such as synergistic effect.
